



**Improving the accuracy in particle concentration measurements of a**
**balloon-borne optical particle counter UCASS**
Sina Jost[1], Ralf Weigel[1], Konrad Kandler[2], Luis Valero[2], Jessica Girdwood[3, 4], Chris Stopford[3],
Warren Stanley[3], Luca K. Eichhorn[1], Christian von Glahn[1], and Holger Tost[1]
[1]Institute for Physics of the Atmosphere, Johannes Gutenberg University, Mainz, Germany
[2]Institute for Applied Geosciences, Technical University Darmstadt, Germany
[3]Particle Instruments & Diagnostics Research Group, University of Hertfordshire, Hatfield,
Hertfordshire, AL10 9AB, United Kingdom
[4]National Centre for Atmospheric Science, School of Earth, Atmospheric and Environmental
Sciences, University of Manchester, Manchester, M13 9PL, United Kingdom
Corresponding authors: Sina Jost (sjost@students.uni-mainz.de) and Ralf Weigel, (weigelr@uni-
mainz.de)

## 1 Abstract

For balloon-borne detection of aerosols and cloud droplets (diameter $0.4 < D_p < 40$ μm), a passive-
flow Universal Cloud and Aerosol Sounding System (UCASS) was used, whose sample flow rate is
conventionally derived from GPS-based balloon's ascent rates. Improvements are achieved by
implementing thermal flow sensors (TFS) 94 mm downstream of the UCASS detection region for
continuously measuring true UCASS sample flow velocities. UCASS-mounted TFS were calibrated
during wind tunnel experiments at up to 10 m s⁻¹ also under various angles-of-attack (AOA), as
these vary during actual balloon ascents. It was found that the TFS-calibration is determined with
sufficient precision using three calibration points at tunnel flows of ∼ 2, 5, and 8 m s⁻¹, simplifying
efficient TFS-upgrades of numerous UCASS. In iso-axial alignment, UCASS flows are accelerated
(by ∼ 11.3 %) compared to tunnel flows (at 2 – 8 m s⁻¹). In-flight comparisons revealed that
UCASS sample flows rarely match the balloon's ascent rate, instead, equality ($v_{GPS} = v_{TFS}$) is
achieved only at AOA ≠ 0°, potentially affecting the UCASS-internal flow pattern and particle
transmission efficiency. To minimise errors on calculated UCASS-based particle number
concentrations, real-time measurements of the true UCASS flow velocity are recommended.



## 2 Introduction

Weather balloon soundings are an established method to carry out in-situ measurements up to high altitudes in the atmosphere (e.g. Golden et al. (1986), Vömel and Fujiwara (2021), or Vömel and Ingleby (2023)). Balloons filled with helium or hydrogen can reach heights of up to 40 km. Due to a moderately variable ascent rate of $\sim 5$ - $6$ m s$^{-1}$ on average, the troposphere and stratosphere are probed with a higher vertical resolution compared to aircraft measurement due to the higher climb and descent rates of the latter (at least $6$ m s$^{-1}$ and usually more, cf. Eufar (2000)). In conventional operation, a radiosonde is used as payload deployed on a thin cord of about 50-60 m length below the ascending weather balloon. Radiosondes are used to directly measure atmospheric pressure, humidity, temperature, and wind parameters during the ascent, while the data are sent in real time via radio to the receiving station (see, e.g., Dirksen et al. (2020)).

In this way, the current state of the atmosphere is profiled, which allows for deriving atmospheric characteristics such as, e.g., its stability or stratification. The payload of the weather balloon can be extended beyond the sole use of radiosondes, for example by including an ozone probe (cf. Smit et al. (2024) and references therein) and/or optical particle counters (Matsumura et al. (2001); Kasai et al. (2003); Smith et al. (2019); Kezoudi et al. (2021); Snels et al. (2021)). Both ozone, as an atmospheric trace gas, as well as aerosols and clouds have a major influence on the Earth system. Aerosols and clouds influence the Earth's energy balance (by absorbing and scattering short-wave radiation), the global water cycle (through the formation of precipitation) and the atmospheric dynamics (through the conversion of latent heat in the phase transitions of water). Considering, that aerosols and cloud droplets modify radiative fluxes (see, e.g., Stier et al. (2024) or Ipcc (2023)), a better understanding of their size, number and vertical distribution is required and the vertical distribution of aerosols and cloud droplets need to be frequently investigated. The University of Hertfordshire (UK) has developed an optical particle spectrometer for balloon soundings: The "Universal Cloud and Aerosol Sounding System" (UCASS, see Smith et al. (2019); Kezoudi et al. (2021); Girdwood et al. (2022); Girdwood (2023), Schön et al. (2024)) is used to detect aerosol particles and cloud droplets with diameters of 0.4 to 40 μm. The probe's measuring principle is described in detail in Sect. 3.

In general, the thought model underlies the balloon sounding, that the horizontal wind that drifts the balloon or influences its geographical position has almost no or a negligible influence in the inertial system of the balloon itself, unless gusty winds or strong wind shear prevail. Hence, regarding the horizontal wind component, there is almost no relative horizontal wind aboard the balloon or its payload. The predominant flow relative to the balloon and its payload that mainly drives the air flow through a UCASS is the vertical wind component, which in our case is determined by the balloon's lift, i.e. its ascent rate.

To calculate a particle number concentration from the particle number detected (per unit time) by a balloon-borne UCASS, the sample volume flow rate through the UCASS detection region must be known. Conventionally, the sample volume flow was determined using the GPS-measured ascent rates $v_{GPS}$ (Kezoudi et al., 2021). This approach has two major disadvantages: a) a zero horizontal wind velocity is inherently assumed, and b) changes in airflow caused by the instrument housing, or small fluctuations remain unconsidered. One goal of this work therefore is to modify a UCASS such that a thermal flow sensor (TFS) is used to continuously determine the flow rates through a balloon-borne UCASS. This enables the accurate and precise determination of particle concentrations (and inferred microphysical quantities) at a higher confidence level.





Beyond that, this study is a prerequisite for investigations based on UCASS measurements during
our balloon missions in Central Europe during summer of the years 2023 and 2024. Ultimately,
the proposed improvements may provide support for UCASS applications by other users.
The present publication is structured as follows. First, the instrument is described. This is
followed by reports on the testing and characterization of the method of mechanically coupling
the TFS to the UCASS (Sect. 3.5.1). Subsequently, the experiments aiming at the optimum position
of the TFS within the flow through a UCASS are described (Sect. 3.5.2). The TFS are then calibrated
against a reference flow sensor (Prandtl pitot tube, PPt) within the flow through a UCASS (Sect.
3.5.3). Based on a numerical simulation, it has already been assumed that the flow velocity within
the iso-axially aligned UCASS would increase by 12 % as compared to the ambient air flow speed
of 5 m s$^{-1}$ (Smith et al., 2019). Therefore, in this work the PPt-measured ambient flow velocities
and TFS-detected flow velocity inside the UCASS are compared (Sect. 4.2). The impact of a non iso-
axial alignment (i.e. when the angle of attack, AOA, towards the UCASS varies due to the balloon-
payload's pendulum motion) on the flow velocities through the UCASS and on the TFS calibration
curves are investigated (Sect. 4.3 and 4.4). Finally, balloon soundings with the UCASS - TFS
combination are used to prove the TFS performance under atmospheric conditions (Sect. 5). The
findings are summarised in the conclusion section.

## 3 Instruments and methods

### 3.1 Universal Cloud and Aerosol Sounding System (UCASS)

The UCASS is an optical particle spectrometer (Smith et al. (2019), Girdwood (2023)). Due to its
small size, low weight and comparatively low cost (approx. 2700 € per unit), the UCASS is well
suitable for balloon soundings. The UCASS is of tubular shape with 183 mm in length and with an
outer diameter of 64 mm at a weight of about 280 g. A hollow flow tube with quasi-elliptical cross-
section of 40 mm × 30 mm diameter extends along the longitudinal axis of the UCASS, through
which the sample air flows and in which the particles are detected. Particle detection with UCASS
is carried out by optical detection of the scattered light caused by particles upon crossing a diode-
emitted laser beam. The optical system of the UCASS defines a particle detection region of 0.5 mm$^2$
in size (Girdwood, 2023). In Fig. 1, the principal setup of the UCASS instrument is illustrated, for
more details see Smith et al. (2019), Girdwood et al. (2022), or Girdwood (2023).
Particles are counted, which pass through the instrument's optically sensitive region along the
detection laser. The ratio of the scattered light generated by a penetrating particle is determined
by means of two annular elements in the detector optics. This signal ratio is judged according to
whether a detection event contributes an accepted count value or not. Based on scattered light
intensity, the accepted counts are categorised into size classes (16 bins) using size-based look-up
tables obtained by calibrations and based on Mie theory. Depending on the chemical composition
of detected particles, different refractive indices must be considered, for example between 1.31+0j
(water) and 1.52+0.002j (Saharan dust) (see Girdwood (2023) for further, more specific details)
whilst for Saharan dust also other values are found, e.g. 1.53−0.0015j (Kandler et al., 2007).
Knowledge of the flow velocity and the dimensions of the sensing volume is required to
determining accurate particle concentrations.

### 3.2 The thermal flow sensor (TFS)

The thermal flow sensor (TFS) model "FLW-122" for the gaseous media by B+B Thermo-Technik
GmbH was used for this study. The TFS has a length of 6.9 mm, is 2.4 mm wide, and 0.2 mm high





(without electrical connections). The TFS surface consists of two platinum resistor elements, one
of which is the heater and the other is the reference element. The heater is a small-sized low-
resistance element ($R_H(0\ °C) = 45\ \Omega$), the reference element ($R_S(0\ °C) = 1200\ \Omega$) is of high
electrical resistance. The two platinum resistors are interconnected and are adjusted via applied
voltage to a specified temperature difference. The heat dissipation from the heater to the
environment corresponds to the energy loss per unit of time, which in turn corresponds to the
power $P$ converted in the resistor.
The temperature of the heating resistor decreases with increasing rate of flow surrounding the
TFS. As platinum is a PTC thermistor, the conductivity of the resistor increases as the temperature
falls, hence the resistance of the heater decreases. To counteract this and to maintain the
differential temperature between heater and reference, the voltage is increased. The higher the
flow velocity, the higher is the voltage (hereafter: signal voltage) required to maintain the
temperature difference. According to the sensor data sheet (B+B-Thermo-Technik, 2016), the
sensor's response sensitivity is 0.01 m s$^{-1}$ with an accuracy of better than 3 % and a temperature
sensitivity of usually less than 0.1 % K$^{-1}$. The raw TFS data is recorded with a resolution of at least
4 Hz and then averaged to the common 1 Hz data basis. Thus, most important variabilities in the
UCASS-internal flow should be captured in sufficient resolution. In addition, low-frequency
influences such as the pendulum motion (order of magnitude 0.1 Hz) and the rotation (of 1 Hz at
most) of the balloon payload are also temporally resolved. Further detailed information on the
TFS (e.g. concerning power and sensitivity of the implemented Z-diode) may be found in the
manufacturer's manual (B+B-Thermo-Technik, 2016).
For TFS calibrations, the following parameterisation, a modified form of King's law (Guellouz and
Tavoularis, 1995), is often used:

$$U^2 = A + B\ \cdot\ v_{TFS}^N\,, \qquad\qquad 1$$

where $A$ (in V$^2$), $B$ (in V$^2$ s m$^{-1}$), and $N$ (dimensionless) are coefficients determined by calibrations,
$v_{TFS}$ is the flow velocity (in m s$^{-1}$) and $U$ is the electrical output voltage (V). Rearrangement of Eq. 1
leads to the desired expression for the flow velocity (here $v_{TFS}$) as a function of measured voltage.
King's law, a combination of Nusselt number and Reynolds number (see Cardell (1993) or
Bearman (1971)), is primarily influenced by temperature fluctuations due to the temperature
dependence of air's thermal conductivity and the dependence of the kinematic viscosity on the air
density. Extensions of this equation by correction terms allow for considering changes of ambient
temperature (see Cimbala and Park (1990)). On the one hand, these corrections apply to
significantly warmer temperatures (300 – 307 K) than relevant for balloon soundings, and on the
other hand require a set of temperature measurements, including the accurate temperature of the
TFS surface itself. The TFS calibrations to assign a flow velocity to the output signal were carried
out under laboratory conditions. To account for the dependence of the in-flight flow measurement
on air density, a corresponding correction (dependent on static air pressures and ambient
temperatures upon vertical sounding) is applied to measured data during post-flight analyses.
Based on the ideal gas law and on laboratory conditions during calibrations ($p_0$, $T_0$, and humidity
expressed as virtual temperature), the correction $v_{TFS}^{corr}$ is obtained by multiplying $v_{TFS}$ (according
to Eq. 1) with a correction factor (at given ambient conditions $p_{amb}$, $T_{amb}$):

$$\boldsymbol{v}_{TFS}^{corr} =\ \boldsymbol{v}_{TFS}\ \frac{\boldsymbol{p_0}}{\boldsymbol{p}_{amb}}\ \cdot \frac{\boldsymbol{T}_{vi}}{\boldsymbol{T_0}} \qquad\qquad 2$$





The virtual temperature $T_{vi}$ was used instead of the ambient temperature $T_{amb}$. Hence, any change
in air's molar weight, air's mass density, and heat capacity due to water vapor load is accounted
for. As long as no phase conversion occurs the temperature ratio also reflects the current moisture
conditions in the atmosphere.
In addition, a cold chamber test was carried out at various temperatures down to – 20°C
(electronic supplement, S1) to investigate the sensitivity of the TFS system (particularly its
electronic compounds) to significantly colder temperatures than those under laboratory
conditions.
To mount the TFS on the UCASS, a 3D-printed polylactide case was produced. This was precisely
adapted to the dimensions of the inner flow tube and the outer shape of the UCASS and extends
the total length of the UCASS by a further 49 mm (cf. Fig. 2).

### 3.3 The Prandtl-Pitot tube

The Prandtl-Pitot tube (PPt) model "TSI 8710 plus" (re-calibrated in July 2022 for flow velocities
of 1 – 10 m s$^{-1}$) by TSI Inc. was used as reference instrument for calibrating the TFS. Via the pitot
tube's entry, aligned against the direction of flow, the stagnation pressure (the sum of the dynamic
and static pressure components) is measured. The outer tube of the Prandtl-instrument with ring-
shaped perforation, allows for measuring the static pressure component. Conversion of
Bernoulli's theorem to resolve to the flow velocity, leads to:

$$v_{\text{PPt}} = \sqrt{\frac{2(p - p_{\text{stat}})}{\rho}}, \qquad\qquad 3$$

where $p_{stat}$ is the static pressure (in hPa), $\rho$ is air's mass density (in kg m$^{-3}$), $v_{PPt}$ is the flow velocity
(in m s$^{-1}$), and $p$ is the stagnation pressure (in hPa). According to the PPt's calibration certificate
an uncertainty of generally less than 1.5 % (at 1 m s$^{-1}$) is to be expected, which decreases to
~ 0.6 % at a flow velocity of 10 m s$^{-1}$.

### 3.4 The wind tunnel

The calibrations have been conducted in the horizontal wind tunnel of the JGU Mainz. This facility
(schematic in the electronic supplement, Fig. S 3) has a total length of 4.6 m and a circular outlet
opening of 0.64 m. The maximum achievable wind speed is approximately 20 m s$^{-1}$ generated by
a 12-bladed impeller rotor in combination with a 14-bladed stator, both of which have an outer
diameter of 0.8 m. The rotor hub (0.4 m in diameter) is encased by a tapered cone downstream of
the stator to aerodynamically optimise the transition from the impeller housing (total length
1.25 m) to the flow channel (3.41 m length). The flow channel widens at 7° towards the horizontal
over a horizontal distance of ~ 2.0 m. In the area of the maximum channel diameter, a gauze, and
a honeycomb mesh with a total thickness of 0.1 m are installed to laminarise the flow. Further
downstream, the flow channel diameter reduces via a radiused narrowing to compress the air
flow. The channel's outlet with a constant diameter of 0.64 m extends over a distance of 0.35 m.

### 3.5 Calibration setup and procedures

Generally, all setups for the TFS calibration were placed centrally within the wind tunnel's laminar
exit flow and in line with the tunnel's ring outlet edge to prevent potentially forming turbulence
from affecting the calibration.



### 3.5.1 TFS – PPt interactions

For exploring whether an extension of the UCASS by the TFS housing (with installed TFS) causes
a measurable impact on the flow at the position of the UCASS optical detection region, a 3D-
printed replica of the UCASS housing was used. Holes were drilled into the housing replica at two
different positions along its longitudinal axis in order to 1) place the PPt inlet at the position of
the optical detection area within the flow tube and 2) place the PPt inlet inside the TFS housing at
the TFS's position while replacing the TFS. The replica housing was lacking all optical elements
(laser diode, mirrors or photodiodes) that an operational UCASS comprises; all bulges along the
flow tube's inner wall were evened out for this experiment.
Furthermore, a thermal anemometer (TA, by TSI Inc., model TSI 8455-300-1) was used to control
the ambient flow generated by the wind tunnel. The TA was calibrated against the PPt in the free
tunnel flow at flow velocities of 2 – 8 m s$^{-1}$. Then the experimental setup of the UCASS replica with
integrated PPt was aligned iso-axially in the central flow of the wind tunnel. The experiments
carried out are listed in Table 1.
The experiments were conducted at ambient wind tunnel air flows of $2 \leq v < 10$ m s$^{-1}$ in
increments of $\sim 1$ m s$^{-1}$. Ten individual measurements were averaged for each of the seven
different wind speeds. The result of this experiments is shown in Fig. S 4 as correlations between
TA-measured flow speeds ($v_{TA}$) against those from the PPt ($v_{PPt}$) within the UCASS flow tube.
Notably, the UCASS tube flow velocity was generally increased compared to ambient wind speeds
if the UCASS is iso-axially aligned with the ambient flow field, which was previously described by
Smith et al. (2019) and confirmed during calibrations presented herein (see Sect. 4.2).
This experiment series demonstrate that applied modifications (extension of the flow tube
geometry and implementation of the TFS) have hardly any measurable influence on the flow
through a UCASS. At balloon ascent velocities, usually well below 10 m s$^{-1}$, none of the data series
stood out from the statistical uncertainty. Therefore, the technical modification proposed herein
is expected to have negligible impact on the measurement performance of the UCASS.

### 3.5.2 Cross sectional flow profile

For the following, a TFS housing shell without installed TFS was mounted on a UCASS to exclude
any obstacle for the air stream through the flow tube. The PPt was attached near the outlet of the
TFS housing so that the PPt inlet was iso-axially aligned and positioned as far as possible inside
the flow tube along the longitudinal axis of the UCASS. Thus, along the setup's longitudinal axis,
the PPt inlet was located at the point where the TFS would normally reside. The PPt was moved
perpendicularly to the flow and in small increments ($\sim$1-2 mm) along a line between the two
extremes of the flow tube's elliptical cross-section. The flow speed in the wind tunnel was kept
constant, and for a single data point the average of fifteen velocity measurements was taken at
each PPt position. The profile measurements of the tube flow profile were carried out for a tunnel
wind speed of about 5 m s$^{-1}$ and 7 m s$^{-1}$.
Figure 3 shows corresponding results where the horizontal bar indicates the maximum of the
standard deviation $\sigma$ of the PPt data obtained from this measurement series. The position where
the TFS would be aligned (if - except for this experiment – installed and adjusted by means of a
positioning device) is marked by the grey-shaded area in Fig. 3. The flow profiles exhibit
asymmetry. The boundary layer at the positive end of the y-axis seems broader than at its negative
end. The reason for this is most likely the quasi-elliptic but asymmetric flow tube cross section of
the UCASS flow tube. As the ambient flow increases, an increased boundary layer thickness is



visible in the UCASS internal flow profile. Moreover, the flattening of the flow profile is clearly
visible towards the centre of the flow tube. For typical UCASS flow rates (here ~ 2-10 m s$^{-1}$) this
demonstrates, that at intended installation position (grey shaded area), the TFS flow
measurement occurs in the free tube flow of the UCASS. Notably, the determined cross-sectional
profile applies to a straight-line flow (i.e. the AOA equals zero). Under variable AOA (see Sect.
3.5.4) and at the selected TFS position any wall effects still have the least influence on the TFS
measurement compared to other positions along the tube's cross-section.

### 3.5.3 Iso-axial flow calibrations

With the iso-axially aligned setup (see beginning of Sect. 3.5) and with the PPt inserted from the
rear into the UCASS/TFS housing, such that the PPt inlet is positioned close to the TFS (cf. 3.5.2
and Fig. 2b), the flow calibrations were performed. The TA installed in the free wind tunnel flow
(outside the UCASS housing, see 3.5.1) was used to control the ambient flow speed.
The TFS calibration comprised a series of measurements at different flow velocities. From slightly
more than 0 m s$^{-1}$ stepwise increasing flow velocities were set in the wind tunnel until ~ 10 m s$^{-1}$
was reached. About 25 measured values were recorded for the calibration of a single TFS. As
disturbance-related fluctuations occurred in the measured values during the calibration of both
the PPt and the TFS, fifteen measured values were recorded for each of the sensors at each
ambient flow speed set. Mean values and standard deviations were determined from these PPt
and TFS data. Each TFS was calibrated individually, and the results of these calibration series are
summarised in Sect. 4.1 ff. After changing the TFS, the iso-axial and centred alignment of the
experiment setup at the wind tunnel outlet was frequently checked and readjusted if necessary.

### 3.5.4 Flow calibrations under variable AOA

For balloon soundings, the payload (incl. UCASS) is usually attached approx. 60 metres (via
unwinder, model "UW1" by Graw GmbH & Co. KG) below the balloon on a cord. The long cord is
intended to dampen the pendulum motion of the payload. Due to the vertical offset between the
balloon and the payload, also a horizontal offset between both ballon and payload occurs
depending on the horizontal wind speed. The balloon may lead the payload by a certain horizontal
distance, unless the horizontal wind speed is zero. Hence, during the balloon's ascent and with
horizontal winds the UCASS is likely inclined at a certain angle relative to the direction of vertical
lift. In addition, pendulum motion and rotations of the payload during flight cannot be completely
prevented, which can lead to a variable alignment of the UCASS relative to a straight ascent due to
several superimposed effects. Therefore, the impact of inclinations of the UCASS with respect to
the direction of flow on 1) the TFS calibration and 2) on the flow velocity within the UCASS is to
be quantified.
The experimental setup was aligned and positioned as described previously (beginning of Sect.
3.5). The iso-axial alignment corresponds to the zero position ($\varphi = 0°$, $\vartheta = 0°$). As the UCASS flow
tube is mirror-symmetrical along the vertical axis, the UCASS was horizontally inclined only along
the positive x-axis ($\varphi$) for changing AOA. Due to the quasi-elliptical shape of the UCASS flow tube,
the effects of the flow angle in y-direction ($\vartheta$) were analysed for both the positive (UCASS inlet
points upwards) and the negative (UCASS inlet points downwards) deviation from zero. 4
illustrates the measurement setup and provides an example of the angular distribution under
which the UCASS was aligned relative to the ambient flow field. Note, that the frame for holding
the UCASS/TFS setup also fixes the PPt (marked in Fig. 4a), which remains aligned with the UCASS
longitudinal axis and is positioned as close as possible at the TFS. Any angular change in the flow
direction to the UCASS inlet therefore affects the entire assembly (cf. Fig. 4c). At the beginning of





each measurement series, data were recorded for five different ambient flow speeds in the zero
position (0°, 0°) to verify the calibration curve of respective TFS used. The angles were then set
within the angle grid shown in Fig. 4b. Fifteen measured values of the PPt and the TFS were
recorded for five different tunnel flow speeds between about 3 and 7 m s$^{-1}$ and the mean value
with standard deviation were determined from PPt and TFS data. The results are summarised in
Sect. 4.3. An attempt was made to reproduce the set tunnel flow velocities for each measurement
series at varied AOA, which allowed also for investigating the relationship between the flow
velocities outside and inside the UCASS housing at variable AOA (see Sect. 4.4).

## 3.6 UCASS internal flow estimates

Besides the counted and classified particles, UCASS also records the particles' average beam
transit times (named MToF, Mean Time of Flight) in four different size ranges. In principle, these
beam transit times could be used to estimate the flow speed in the measurement volume and
therefore would provide another option for a sampling volume assessment. Such an approach is
used for example in connection with the particle detectors by Alphasense OPC-N2 and OPC-N3
(Bezantakos et al., 2020). We therefore correlated the measured TFS velocities with the inverse
of the different MToF values reported for each interval, when particles were detected. A
considerable number of apparently clipped values (piling at the apparent bottom and top end of
the measurement scale) were removed. Still, while there was a very weak correlation found
between the inverse MToF and the TFS, the huge variations, the clipped values and the scarcity of
data - in most 0.5 second intervals no particles are detected - prevent this approach from being
used as a measurement for a flow rate in UCASS.

## 4 Calibration results

To this point, it has been demonstrated that

310        1) the insertion of the TFS into the flow tube of the UCASS has no measurable impact on its
311           particle-sensitive area (Sect. 3.5.1) and
312        2) as the TFS installation is achieved by using a positioning device it is systematic for all TFSs
313           used whose measurements occur outside the tube's boundary layer in the central UCASS
314           flow.

Hence, the assumption seems appropriate that integrating TFS into the UCASS flow tube did not
generate any additional interferences for the calibrations discussed hereinafter.

## 4.1  High resolution (HRC) and three-point calibration (TPC) of the TFS

According to the functional relationship described in Eq. 1 the measured TFS output voltage is
calibrated versus the PPt-measured flow velocity, which is used hereafter as the calibration curve
of a TFS (further details in Sect. S 2 in the electronic supplement). For each measuring point, the
mean value plus (minus) the associated standard deviation - i.e. the extremes of variability - was
inserted into the calibration curve function yielding a deviation from the averaged measurement
point by not more than 0.08 m s$^{-1}$ corresponding to a percentage maximum of 1.3 %. The results
show that the different TFS exhibit very similar behaviour and that the parameterisation of the
calibration curve function is valid for all calibrated TFS. Table S 1 (see electronic supplement) lists
the parameterisation coefficients for each TFS. Repeated calibration series with a larger time
offset and with the same TFS after disassembly/reassembly of the setup also reproduced previous
values within 1.8 % at maximum, and on average within 1 %, in the relevant range of ambient
flows. However, the measurements of the calibration curves were likely influenced by the ambient





temperature and pressure conditions in the laboratory as both affect air's mass density, which in
turn controls the heat advection at the TFS.
In the process of these relatively high-resolution calibrations (HRC) and analyses of the
calibration curves, it was found that only three calibration points reproduce the calibration curve
with acceptable accuracy (see electronic supplement, Sect. S 2, for details). For the TFS used
(TFS 8), and at tunnel flow speeds between 2 and 8 m s$^{-1}$, the three-point calibration (TPC) curves
did not deviate by more than 1 % from the highly resolved initial calibration curve.

## 4.2 Internal versus external flow velocity

The PPt was repositioned next to and slightly upstream of the UCASS inlet to measure in the free
flow of the wind tunnel (setup is shown in Fig. S 5). The UCASS-integrated TFS 8 had previously
been calibrated with the PPt. The UCASS was aligned iso-axially to the ambient flow of the wind
tunnel and centred at the annular outlet of the tunnel. The wind tunnel flows were stepwise
increased from slightly more than 0 to up to 11 m s$^{-1}$ and then decreased with a total of seventeen
different settings. For each of the ambient flow speeds, fifteen velocity values were recorded as
measured by the PPt to be averaged. In Fig. S 6, the PPt-measured flow velocity outside the UCASS
is displayed in reference to the UCASS internal flow velocity determined with the TFS 8. The
velocity range relevant for balloon soundings (2 - 8 m s$^{-1}$) is indicated by the black dashed square.
Within this range, ten different values of ambient flow speed were set for each measurement of
this series. The standard deviation $\sigma$ of the PPt data range between 0.01 and 0.04 m s$^{-1}$. Only the
mean value of the TFS output voltage was noted, which resulted from the automated averaging by
the multimeter used. During the TFS 8 calibrations, standard deviations of at the most 0.08 m s$^{-1}$
were obtained corresponding to a maximum relative deviation of 1.3 % (extremes of variability,
cf. first paragraph in Sect. 4.1). A first-order polynomial fit was created for the recorded
measurement. The resulting line-fit indicates that in an iso-axial arrangement the UCASS internal
flow velocity exceeds the outside flow speed. Calculated from ten data points at external wind
velocities between 2 and 8 m s$^{-1}$, the UCASS internal flow velocity is increased by 10.7 to 11.9 %,
with an average relative deviation of 11.3 %, as compared to the external flow speed.
This result quantitatively compares well with the results of Smith et al. (2019), who simulated an
increase in the UCASS internal flow velocity by 12 % under an iso-axial alignment of the UCASS
and at an ambient flow speed of 5 m s$^{-1}$. Taking into account the standard deviations of averaged
data points determined here may put the small deviation between previous and current results
into perspective.
However, in balloon-borne UCASS field applications, the GPS-based ascent rate is usually used to
determine the sample volume flow and thus the measured particle number concentrations (e.g.
Kezoudi et al. (2021)). So, when a UCASS was iso-axially aligned during a straight balloon ascent
at rates of 2 to 8 m s$^{-1}$, the GPS-based calculation would lead to an underestimated UCASS sample
volume flow by about 11-12 % and the particle number concentrations would be overestimated
accordingly. Therefore, from current point, a direct measurement of the UCASS internal flow
velocity during flight appears useful, but as will be shown (Sect. 5), the conditions during balloon
soundings are more complex and the necessity for a flow velocity detector integrated into the
UCASS gets all the more obvious.



### 4.3 UCASS internal flow velocity under variable AOA

**4.3 UCASS internal flow velocity under variable AOA**
Compared to previous experiment series, the UCASS was intentionally misaligned from the iso-
axial orientation (cf. Sect. 3.5.4).
Figure 5 shows the highly resolved calibration curve TFS 8. The mean values from experiment
series at five ambient flow speeds are also depicted for each AOA. The standard deviations are not
shown for the sake of clarity of the figure. The error in setting the UCASS deflection angle was
estimated to be ±0.5° in each case. In the zero position (0°, 0°), which was calibrated once at the
beginning and at the end of the series, the $\sigma$-values ranged between 0.002 and 0.003 V (TFS 8
output voltage) or between 0.02 and 0.04 m s$^{-1}$ (PPt). With increasing AOA, the standard
deviations ($\sigma$) rose to a maximum of 0.025 V (TFS 8) and 0.09 m s$^{-1}$ (PPt). Hence, the fluctuations
of measured variables generally increased with increasing AOA.
Based on the TFS 8 calibration curve, the flow velocities $v_{\mathrm{TFS}}^{a}(U)$ were determined with the mean
TFS output voltage values ($\overline{U}$). These are compared with the flow velocities $v_{\mathrm{TFS}}^{b}(U)$ resulting from
the extreme values of standard deviation at each mean output voltage (i.e. $\overline{U} \pm \sigma$). The maximum
standard deviation range observed was ± 0.31 m s$^{-1}$ (8.3 %). Oscilloscope records of the variations
in the TFS output voltage over 20 s clearly indicated an increasing scatter intensity when varying
the AOA of UCASS, e.g. from (0°, 0°) to (20°, -40°). The output voltage fluctuations were invariant
in time (in particular non-periodic) and stationary, i.e. rather of a statistical nature.
The mean values of the TFS measurements were inserted into the calibration curve function of
TFS 8 to determine the flow velocities ($v_{\mathrm{TFS}}$) to be juxtaposed to the mean PPt-measured flow
velocities ($v_{\mathrm{PPt}}$). The flow velocity determined with the calibration curve was used as the reference
and an error is considered with
$$\varepsilon = \frac{v_{\mathrm{TFS}} - v_{\mathrm{PPt}}}{v_{\mathrm{TFS}}} \cdot 100, \qquad\qquad 4$$

out of which the maximum percental error and its mean resulted for each AOA. Table 3 shows the
maximum and the mean of the relative percentage deviation of measured values at variable AOA
compared to the calibration curve TFS 8. At AOA of up to 30°, the maximum relative percentage
deviation was 3.5 %, and above 30°, the relative percentage deviation of measured values from
the calibration curve increased significantly. At such large AOA, the $\sigma$ of TFS-measured voltages
also rose substantially. During balloon sounding, UCASS inclinations of more than 30° from the
vertical are not expected. In an earlier study (Smith et al., 2019), a model was used to simulate the
angular inclination of the UCASS in relationship to the direction of ascent. Initial pendulum
oscillations of up to ± 20° were assumed, which were damped to around ± 5° within about
30 seconds. If these conditions occurred at an actual balloon sounding with UCASS, the percentage
relative deviation of measured velocities as determined here (based on the TSF 8 calibration
curve) would be a maximum of 2.8 % under variable AOA smaller than 20°. Hence, even at UCASS
inclinations of up to 20°, the particle concentration is affected by this angle deflection with a
maximum uncertainty of less than 3 % (maximum relative deviation). To validate the TFS 8 data
series, the measurements were repeated with TFS 7 in the same setup. Corresponding results are
summarised in Sect. S 3 (electronic supplement) representing a qualitative and largely
quantitative confirmation of the results and conclusions made.

### 4.4 Internal versus external flow velocity under variable AOA

**4.4 Internal versus external flow velocity under variable AOA**
In Fig. 6, the velocity of TA-measured ambient wind tunnel flows (as reference) is plotted against
the PPt-measured flow velocity inside UCASS. Standard deviations of 0.01 to 0.10 m s$^{-1}$ were





determined for the ambient flow velocity between 2 and 8 m s$^{-1}$. For PPt data, $\sigma$-values ranged
from 0.02 to 0.09 m s$^{-1}$, which increased with rising ambient flow velocities and AOA. With
increasing AOA, the UCASS-internal flow velocity decreased. For AOA of (0°, 30°) and (0°, -30°),
and beyond, the UCASS-internal flow was slower compared to external flow velocities.
The UCASS-internal flows in zero-position (0°, 0°) were used as reference and parameterised by
a first-order polynomial fit. The comparison of a) the calculated flow velocity inside the iso-axially
aligned UCASS with b) its internal flow velocities at AOA of (0°, 30°) and (0°, -30°) revealed a
relative percentage decrease in the flow velocity of around 15 % (see Table 4). For an iso-axially
aligned UCASS, the internal flow is accelerated by on average ∼ 11.3 % (see Sect. 4.2). The
contrary effects of a) a flow acceleration inside the iso-axially aligned UCASS and b) the
deceleration inside the UCASS when deflected from the zero-position are likely superimposed.
Thus, with a UCASS inclination of 30° during a balloon sounding, the internal UCASS flow velocity
is reduced by around 4 % net compared to the external flow. In turn, at such UCASS inclination, a
particle number concentration calculated with the GPS-based ascent velocities underestimates
the prevailing trace material concentrations by only about 4 % under these conditions.
At a certain deflection angle (30°, 0°) a noticeable but undeterminable interference effect appears
to have impacted the laboratory experiments. With increased deflection of the UCASS, the inlet
opening of the UCASS is shifted out of the central wind tunnel flow and towards the tunnel's wall.
With decreasing distance to the wall of the wind tunnel, effects of flow disturbances cannot be
ruled out, which could influence the relative deviation from the calibration curve. Potentially,
signal fluctuations are also amplified by flow disturbances that occur at the leading edge of the
UCASS entrance when the UCASS is deflected too far from the zero-position. The TFS appears to
be sensitive to such fluctuations. In addition, an influence of wind gusts on the experiment cannot
be excluded, as the wind tunnel is operated inside a building, but has connections to the building's
exterior to enable enhanced wind tunnel flows.

## 5 Ambient measurements

Ambient balloon soundings were carried out with two UCASS instruments, such that two TFS were
operated in parallel during respective ascent. The balloons' ascent rate is determined from both
the static pressure measurement ($v_p$) and the GPS data, each of which is recorded independently.
The static pressure in flight, resolved at 1 Hz, is converted to barometric altitudes by the internal
primary data processing in the VAISALA sounding system, as recalculations (based on Sonntag
(1994)) confirm. GPS and pressure-derived ascent rates correlate very well (regression yields a
slope of one and a constant element of - 0.1 m s$^{-1}$ with a coefficient of determination r$^2$ of 0.99).

### 5.1 Vertical profiles

Figure 7 shows vertical profiles of three balloon soundings launched from Tailfingen (Germany),
at ∼ 900 m above sea level (a.s.l.) in August 2023. The profiles allow for comparing 1) the balloon
ascent rate from the GPS measurements $v_{GPS}$, 2) the balloon ascent velocity from the static
pressure measurements $v_p$, and 3) the flow velocities through the two UCASS units (TFS°1 and
TFS°2) operated on each ascent. Results from other balloon soundings are shown in the electronic
supplement (Fig. S 10). The profiles are limited to an altitude of 7.5 km a.s.l., as at about this height
the ambient temperatures fell below the specified temperature limit (253 K, see B+B-Thermo-
Technik (2016)). Further laboratory tests are required to confirm the robustness of the TFS
measurements also at temperatures of 253 - 220 K or to reveal the need of further corrections. As





nearly all UCASS particle measurements were detected below 7.5 km, the restricted height range
nevertheless covers the relevant part of the flight. The data are low-pass filtered by a ten-second
running average.

A common feature of the vertical profiles is the stronger scatter of the GPS-derived ascent rates
compared to those based on pressure measurements. This might be connected to a larger
uncertainty inherent with the GPS. In general, however, both values are well-correlated. Up to
~ 2 km of the balloon sounding of 12 August (Fig. 7a) the TFS measurements agree relatively well
with GPS and the pressure data (see also Fig. 7b). Below~ 3 km, TFS A and TFS B deviate from the
pressure and GPS data by up to 30 %. Up to ~ 3.8 km, the different speed measurements from all
instruments are comparatively consistent. Above 3.8 km, the TFS measurements (again consistent
between TFS A and B) show periodically fluctuating and lower flow speeds (by up to 15 %, and
above 7 km altitude by up to 25 %) compared to those resulting from GPS and pressure.

During the earlier balloon sounding of 16 August (Fig. 7c) and up to an altitude of ~ 2.4 km, the
TFS-based velocities are systematically higher (in peaks by up to 45 %, cf. Fig. 7d) than
corresponding values from GPS and pressure. In contrast to the previous case, the measurements
of the four sensors are quite consistent at altitudes between 2.4 and 5.3 km with deviations of
seldomly more than ± 10 %. Above ~ 5.3 km, still quite consistent TFS measurements increasingly
deviate from the GPS and barometric data with increasing hight and indicate by up to 15 % lower
UCASS flows compared to the external wind velocity.

Later balloon sounding of 16 August (Fig. 7e) shows only at the lowest heights (up to ~ 1.3 km)
more or less consistence of the four measurements. Between 1.3 and 2.7 km, the TFS- velocities
are generally higher (on average by 15 %, in peaks up to 25 %, cf. Fig. 7f) than those derived from
GPS and pressure. Above 3.2 km, reversely the ascent velocities from pressure and GPS slightly
exceed the TFS data. At altitudes of 4.5-5.5 km and above ~ 6.2 km, GPS and pressure
measurements commonly reveal increased vertical velocities that rise from about 6 m s$^{-1}$ to about
8 m s$^{-1}$. Within these layers TFS-velocities are mostly 15 % (in peaks up to 25 %) slower than the
ambient air flow. The origin of these features in the GPS and pressure-based velocity profile
apparently has, if any, only minor impact on the TFS measurements.

These examples suggest that over large parts of the vertical sounding (up to 7.5 km altitude) the
flow velocity within the UCASS (and thus through its optically sensitive particle detection region)
does not match the GPS or $p$-derived ambient flow velocity.

## 5.2 Implications of observed features

Influences on the UCASS-internal flow speed and pattern due to an additional installation of a TFS
in a housing extension, if any, are negligible (Sect. 3.5.1). The sensitive area of the implemented
TFS is located within the free tube flow through a UCASS, i.e. outside the boundary layer of the
UCASS flow tube's inner walls (Sect. 3.5.2). Moreover, with an iso-axial alignment of the UCASS,
the inner flow is accelerated by ~ 11.3 % when compared to the ambient flow velocity around the
UCASS (Sect. 4.2), which is in general agreement with earlier findings (Smith et al., 2019). As
discussed in Sect. 4.4, with increasing angular deflection of the UCASS from the iso-axial
orientation, the flow velocity inside a UCASS is decelerated towards values of the ambient flow
speed and below.

The payload as used during the field mission had a total weight of up to 3.9 kg. About 10,000 litres
of helium are required for the balloon to yield enough buoyancy for desired lift. Under conditions



with horizontal winds, the effective surface area of the filled balloon is many times larger than that of the payload structure (<0.1 m³). When fully unwound, the flexible cord between the balloon hull and the payload extends up to ~ 60 m. During ascent from ground through the boundary layer, the balloon (with larger surface area and several tens of meters above the payload) is frequently exposed to different wind speeds due to (variable) wind shear as compared to the payload because horizontal winds increase with increasing height and decreasing ground friction. Therefore, during an ascent under any non-zero horizontal wind speed, this structure will generally be inclined with respect to the vertical as indicated by the horizontal displacement between two radiosondes implied at the opposite ends of one balloon-payload-ensemble of ~ 62 m length in total (see Sect. S 4 in the electronic supplement).

Above the boundary layer, the balloon's much larger surface area may cause it to act as a sail in prevailing horizontal winds, causing the payload with a much smaller surface area to be towed behind. Therefore, the balloon is most times slightly dislocated with respect to the resultant wind vector compared to the payload, the payload will inevitably be inclined relative to the wind vector over large parts of the balloon sounding.

Based on the ratios of the TFS measurements compared to the ascent velocities from GPS and pressure measurements, different scenarios arise over a flight, in which

1. The flow velocities through the UCASS are increased compared to the ascent velocities: as if the deflection angle of the UCASS relative to the vertical was comparatively small or close to an iso-axial alignment of the UCASS within the ambient flow,
2. The flow velocities through the UCASS are within the variability range of the GPS and pressure measurements of the ascent velocities: for this, a deflection of the UCASS of about 20°-30° would have to be attained (cf. Fig. 6 and Table 2) while an enhanced uncertainty is hereby accounted for by using the TA instead of the PPt for measuring the wind tunnel flow speed, as the PPt measured at that time inside the UCASS housing,
3. A flow velocity through the UCASS that is smaller than the GPS or pressure-based ascent velocities: for this, a deflection of the UCASS of about >30° would have to be reached.

For the last of these points in particular, the required deflection angles of the UCASS appear extreme to clearly explain observed velocity ratios. In addition, pendulum movements of the payload around a position in the inclined state (i.e. without swinging through the pendulum's equilibrium, which would correspond to an iso-axial, strictly vertical UCASS alignment) could generate a continuously variable orientation of the UCASS in the flow. Moreover, it was observed that the payload can rotate around its vertical axis (in line with the balloon cord). The superposition of these movements and others resulting from the variability of the payload's lift (e.g. indicated by fluctuations in GPS and pressure-based ascent rates) may lead to various influences on the flow through the UCASS, which currently are neither separable nor quantifiable. As the ascent speed is mainly determined by the balloon's buoyancy, drag force, and gravitation, the resulting wind situation at the payload may differ from that at the balloon due to small scale atmospheric variability (e.g. turbulence).

During balloon soundings, the flow velocity through the UCASS is subject to complex influences and is largely decoupled from the pure vertical velocity (i.e. the ascent rate) of the balloon. The high variability of the flow velocity measured during ascent directly inside the UCASS emphasises the importance of such a direct measurement. The frequent deviation of the UCASS flow speed from the GPS or pressure-based vertical velocity is one further argument in favour of





implementing an independent flow sensor inside the UCASS. The GPS or pressure-based ascent
rates deviate by values in the range of metres per second from the UCASS internal flow velocity.
Based on a velocity of 6 m s⁻¹, a deviation by ± 0.5-2 m s⁻¹ results in an error of ± 10-30 % and a
corresponding over- or underestimation of the resulting particle number concentration.
Table 5 summarises the main uncertainties arising from the different methods for determining a
flow velocity through UCASS and which have equal quantitative impact on resulting particle
number concentrations. From the Gaussian error propagation of the relationship given in Eq. 1
and Eq. 2 an overall uncertainty $\Delta\,v_{\mathrm{TFS}}^{\mathrm{corr}}$ combines from 1) the uncertainties in $T$ and $p$ measured
with a radiosonde RS 41 SGP (Vaisala, 2024) also including uncertainties arising from the TFS
calibrations conditions (in $p_0$ and $T_0$) and 2) the uncertainty $\Delta\,v_{\mathrm{TFS}}$ (i.e. determined $\sigma$) obtained
from the cold chamber test at different ambient temperature conditions (cf. Sect. S 1 in the
electronic supplement). For three chosen temperatures (i.e. 275 K, 264 K, and 254 K) from these
chamber experiments the corresponding deviations of the GPS-based flow velocities ($\Delta\,v_{\mathrm{GPS}}$) from
corresponding $v_{\mathrm{TFS}}^{\mathrm{corr}}$ are determined at equal temperature conditions during three vertical
soundings (Fig. 7 and adjacent text). Thus, the total uncertainty in $v_{\mathrm{TFS}}^{\mathrm{corr}}$ is fairly consistent within
a range between 6.9 % and 8.8 % (values marked in grey in Table 5). In a few of the conditions
selected here (275 K and 264 K), resulting $\Delta\,v_{\mathrm{GPS}}$ is comparatively low (values framed between
two vertical lines in Table 5), or in almost similar range of $\Delta\,v_{\mathrm{TFS}}^{\mathrm{corr}}$ (values framed between two
horizontal lines) however, in other cases (enclosed framed values) the offset in $v_{\mathrm{GPS}}$ is significant
and can exceed 30 % (occasionally even more as seen in Fig. 7d). In essence, $v_{\mathrm{GPS}}$ appears as the
more precise measurement, but can unpredictably become highly inaccurate upon vertical
sounding over large height ranges (500 - 1500 m). In contrast, $v_{\mathrm{TFS}}^{\mathrm{corr}}$ presumably is the more
accurate measurement over discussed vertical range up to ~ 7.5 km, with a comparatively higher
though determinable imprecision.

## 6 Summary and Conclusions

This study indicates a possible improvement of the Universal Cloud and Aerosol Sounding
System (UCASS), a balloon-borne optical particle counter, to obtain aerosol and cloud droplet
concentration measurements with increased accuracy. The integration of a thermal flow sensor
(TFS) into the UCASS allows for direct, real-time and continuous flow velocity measurements
in the immediate vicinity of UCASS's particle detection region. This approach resolves
inaccuracies arising from the conventional reliance on GPS or pressure-based balloon ascent
velocities, which do not account for gusts or strong wind shear, payload oscillations, flow
distortions (caused by the UCASS housing), and particularly the deflection of the UCASS body
from an iso-axial alignment during ascent.
Wind tunnel experiments showed that the UCASS-internal flow velocity, when iso-axially aligned
with the ambient airflow, is on average 11.3 % faster than the surrounding external flow (between
2 – 8 m s⁻¹), largely consistent with previous investigations. Hence, the airflow dynamics of the
UCASS are reproducible and predictable under controlled conditions. Further laboratory
experiments under variable angles-of-attack (AOA) revealed that with a deflection of the UCASS
by around 20° - 30°, the UCASS-internal flow velocity is reduced to values approximately
corresponding to the external flow velocity.
It was found that high-resolution calibrations of a TFS provide accurate results that are
parameterizable according to King's Law, with deviations of no more than 1.3% from the mean



samples. A more efficiently implementable three-point calibration (TPC) method (using flow
velocities of approximately 2 m s$^{-1}$, 5 m s$^{-1}$, and 8 m s$^{-1}$) showed deviations of less than 2.9 %
compared to detailed calibrations.
During balloon soundings, UCASS-internal flow velocities could deviate significantly from the
GPS-based ascent rates, varying by up to 30 %. Thus, accurate measurements of particle
concentrations based solely on GPS-derived ascent rates are subject to considerable uncertainty.
Vertical profiles from three soundings showed that the deviations in flow velocity increased
with increasing altitude due to changing atmospheric conditions and balloon-payload-geometry
(i.e., the inclination of the payload orientation and thus the UCASS alignment). Referring to
the thought model of the balloon sounding (see Sect. 2), that the flight pattern of the balloon is
driven by the horizontal wind components (i.e. $u$ and $v > 0$): then 1) the balloon body acts as a
sail with its comparatively large surface and 2) the comparatively small payload is towed. In
other words, even if the main wind components with respect to the payload's inertial system
($u_{rel}$, $v_{rel}$) were nearly zero, this will typically result in a constellation of the balloon-payload
geometry which causes a flow towards UCASS with a non-zero AOA. While UCASS is aligned
with the cord to the balloon, it is deflected with respect to the buoyancy-induced vertical lift,
i.e., iso-axial flow to the UCASS is rarely given. The actual flow, which is affected by the
pendulum motion and rotation of the payload and their superposition, is hardly reproducible or
quantifiable. This ultimately underlines the need for a continuous measurement of the actual
internal UCASS flow during the entire flight.
Hence, for airborne particle measurements the calculated particle concentrations from UCASS
detections based on GPS ascent rates may introduce errors of up to 30 %. Real-time TFS data
would reduce this uncertainty by ensuring that the actual air volume of the samples is
determined independently of external conditions. The integration of the TFS into the UCASS
therefore represents an improvement in the methodology of measuring aerosol and cloud
droplet concentrations. By providing direct and precise flow velocity measurements, the TFS
avoids the limitations of GPS-based methods, including errors caused by payload movements,
gusts, wind shear, and flow deviations caused by the shape of the UCASS housing. Field tests
confirmed that this approach leads to more accurate assessments of air volumes in the samples
and can reduce uncertainty in calculating particle number concentrations.
The knowledge gained may motivate future modifications and improvements in the further
development of the balloon-borne UCASS instrument. The robust three-point calibration
methods applied in this study facilitate the technical effort involved in integrating TFS into a
modified UCASS. This work can provide a basis for improved vertical profiling of aerosols and
cloud elements with cost-effective and flexible instruments such as UCASS.
**Code availability**
"scipy.optimise.curve_fit" ((Scipy-Community, 2023))
**Data availability**
to be announce
**Author contribution**
SJ performed the calibrations, reporting and optimised technical designs. SJ, RW, KK wrote the
article with contributions by JG, CS, WS, LKE, LV and HT. LV, LKE and CvG ensured the technical
operability of the TFS-equipped UCASS instruments (supported by JG, CS, WS) and the balloon
gondola during field missions. HT (forecasting), LKE (preparation and launch) and KK, SJ
(recovery) contributed invaluably to successful balloon soundings.



**Competing interests**
The authors declare that they have no conflict of interest.
**Acknowledgements**
The contributions from the technical staff at the workshops of the Institute for Physics of the
Atmosphere (Mainz University) and of the MPI for Chemistry were crucial and essential. In
particular, we acknowledge the support of H. Rott, K. Wilhelm, M. Maurer, T. Böttger, P. Schumann,
T. Kenntner, M Dietrich, and B. Meckel. We furthermore gratefully acknowledge the excellent
support by R. Dominik, I. Knapp-Wagner, J. Frielingsdorf, M. Euler, and O. Krause as well as the
radiomen (a. o.) W. Hallmann (DF7PN), R. Roth (DG7FDE). We are also deeply grateful for the
generous hospitality of M. & A. Conzelmann (Tailfingen) of A. Lampmann (BSC Spielberg), and the
residents of Spielberg. The authors thank the Editor (EDITORNAME) and (NUMBER) (anonymous
or NAME) reviewers for their careful evaluation of this article and their valuable and constructive
recommendations.
**Financial support**
Our research was funded by the Deutsche Forschungsgemeinschaft (DFG, German Research
Foundation) – TRR 301 – project ID 428312742, in the subproject B02 "BISTUM" within the CRC
entitled "The Tropopause Region in a Changing Atmosphere" ("TPChange"). We also received
financial support by the "Dres. Göbel Climate-foundation".
**Figure captions**
**Figure** 1: Design of the UCASS and arrangement of its optical elements. Figure (a) shows the side
view of the UCASS, while figure (b) shows the front view. The definition of the particle detection
zone (df1) has been added to the figure. Adopted from Smith et al. (2019).
**Figure** 2: (a) The TFS housing ("c1"), which also contains the control and regulation electronics
of the TFS. The sensitive area of the TFS with a length of 6.9 mm is labelled as "c2". (b) The frontal
view of the experimental setup. The perspective is in the direction of flow towards the UCASS front
and into the flow tube, looking at the TFS and the Prandtl-Pitot tube, which is located downstream
behind the TFS.
**Figure** 3: Resulting flow velocity profiles along the elliptical cross-section of the instrument's flow
of the UCASS or of the TSF housing with a diameter of 40 mm × 30 mm (length × width). Along the
cross-section's main axis, the flow velocity measurements were carried out; twice for an ambient
flow velocity of ~ 5 m s$^{-1}$ (forwards "a" and backwards "b", respectively) and once for ~ 7 m s$^{-1}$
(direction "b" only). Each data point represents an average of fifteen velocity measurements. The
maximum standard deviation is given representatively for all data points with indicated
horizontal bar. The position of the TFS (if installed) is marked by the grey area.
**Figure** 4: Measuring setup with UCASS, mounted TFS housing, and installed PPt (from the rear
reaching into the TFS housing), fixed in a vertically and horizontally tiltable apparatus with scales
for reading the deviation angles in each direction a) Side view in horizontal orientation b) Variable
deviation angles, which were set in angular degrees deviating from the zero position (iso-axial
case) during the experiments c) View of the measuring setup under horizontal and vertical
displacement from the zero position.



**Figure** 5: Comparison of the high-resolution calibration curve (HRC) with measurements of the
UCASS-implemented TFS 8 at variable angles of attack. The standard deviations of individual data
points are not shown for the sake of clarity.
**Figure** 6: Comparison of the recorded flow velocities outside ($v_{TA}$) and inside ($v_{PPt}$) the UCASS at
variable angles of attack during the series of measurements with TFS 8. The standard deviations
(at the most ± 0.1 m s⁻¹ at tunnel wind speeds between 2 and 8 m s⁻¹) are not shown for reasons
of clarity. First-order polynomial fits were created for the flow velocities recorded under various
angles of attack to compare with the high-resolution calibration (HRC) at zero position (0°,0°).
**Figure** 7: Vertical profiles of the vertical velocity based on measurements of changed GPS and
barometric altitudes per unit time (i.e. $v_{GPS}$ and $v_p$) and on TFS measurements of the flow velocity
($v_{TFS}^{corr}$) through the UCASS of selected balloon soundings during a field mission a) from 12 August,
launch at 14:10 (LT), c) from 16 August, launch at 11:51 (LT) and e) from 16 August, launch at
13:41 (LT). Ten-seconds-running average of respective data are shown. Additionally, respective
deviations of the differently obtained flow velocities from each other are shown for each vertical
profile (panels b, d, f). Note that the deviation of the ascent rates (i.e. the external flow velocity,
GPS or $p$-based) in reference to the UCASS-internal flow speed differs from zero over large parts
of depicted flight sections.





**Figures**

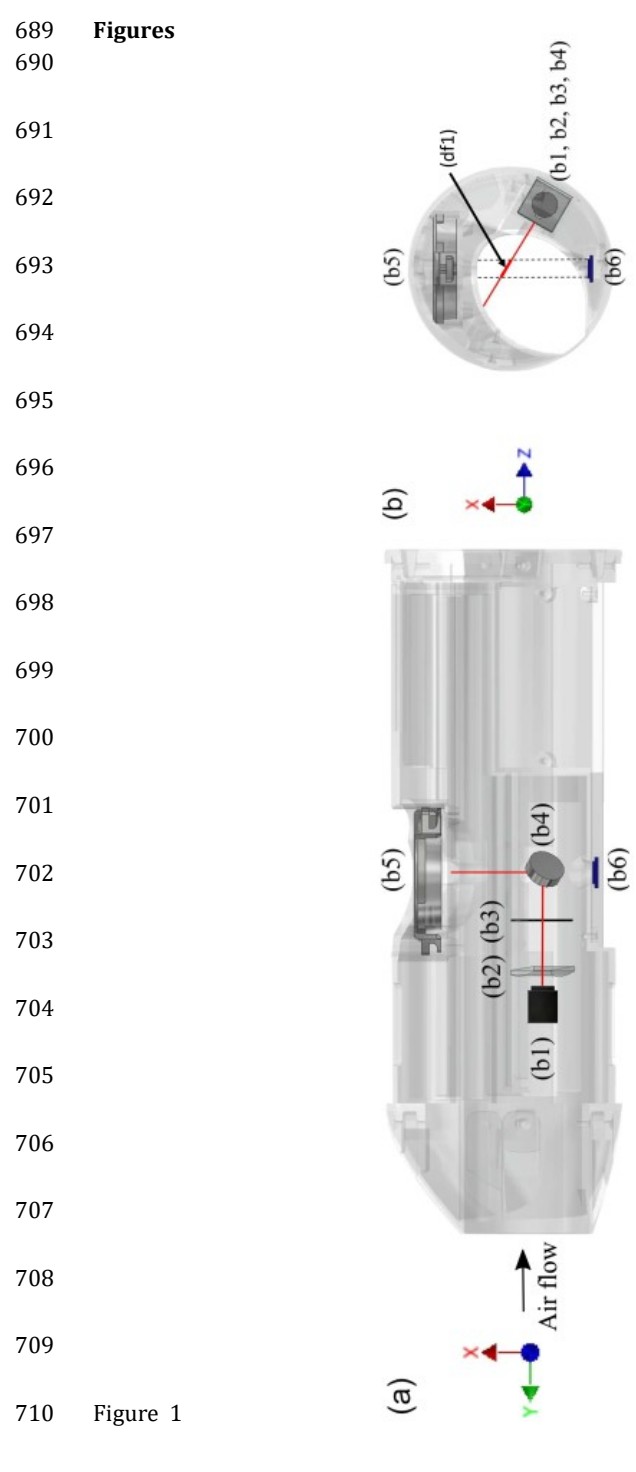

Figure 1


















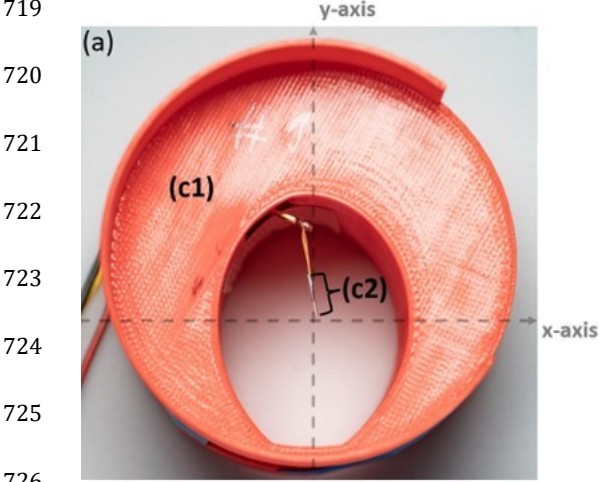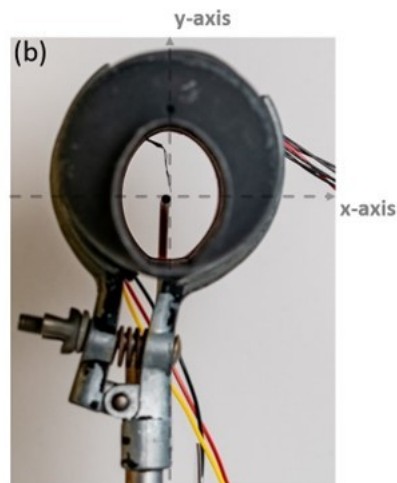

Figure 2



























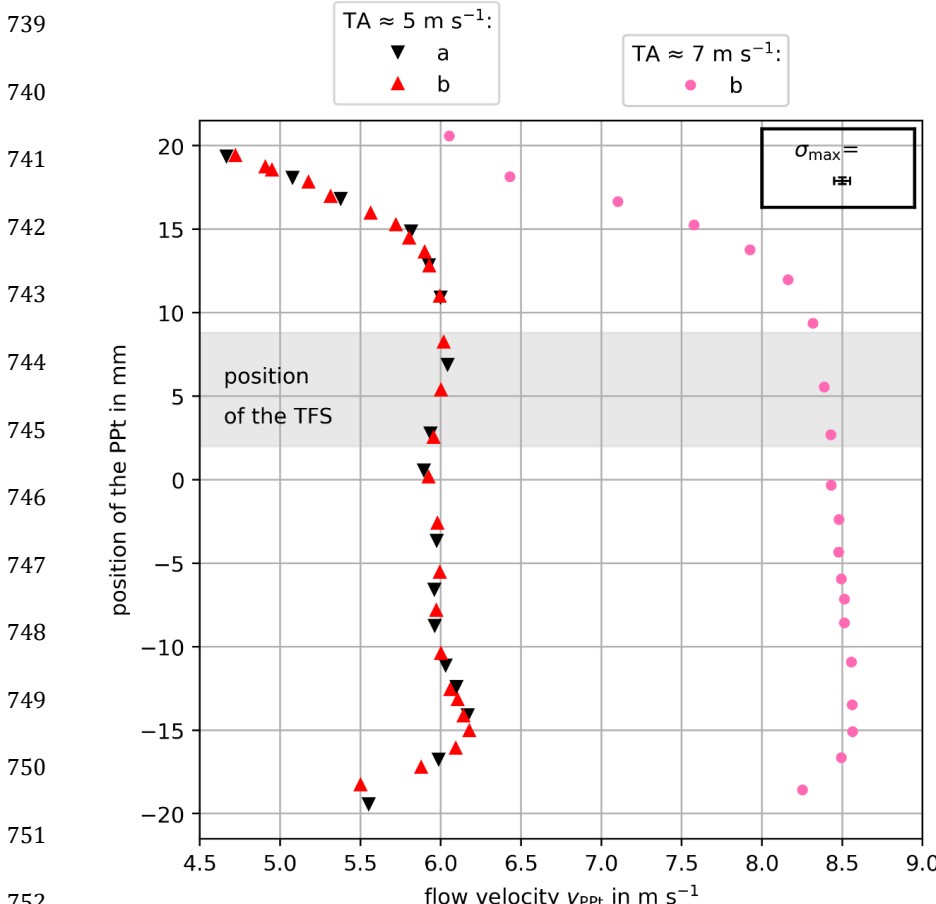

Figure 3



























Figure  4
























Figure 5




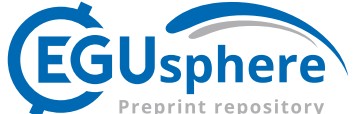

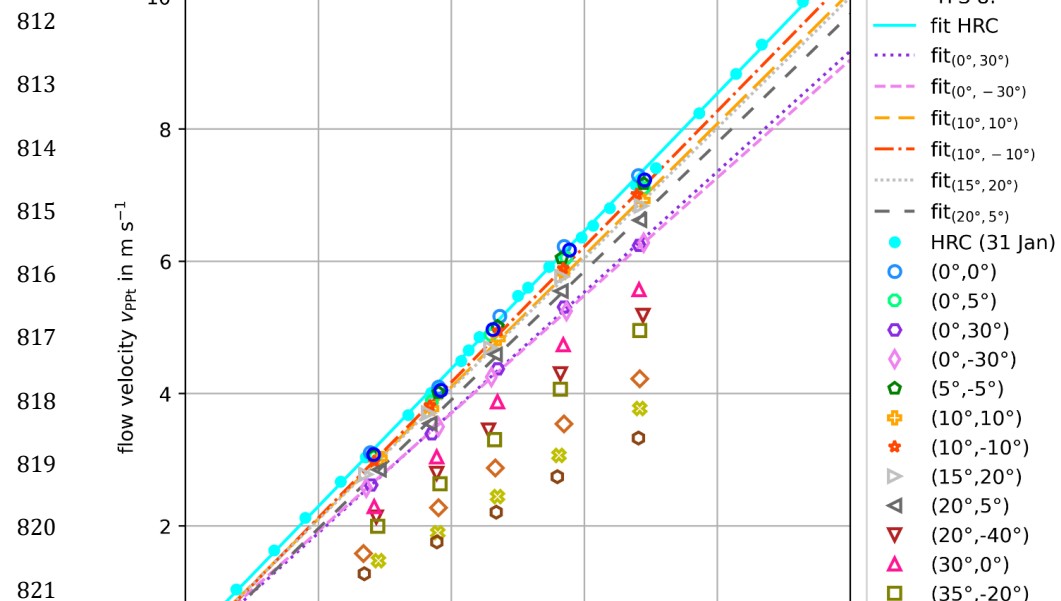

Figure 6



























Figure 7



**Tables**

| Exp. # | PPt | TFS/housing | description |
|--------|-----|-------------|-------------|
| 1 | yes | no/no | PPt in UCASS housing at position of detection region |
| 2 | yes | no/no | "—" |
| 3 | yes | yes/yes | PPt at detection region + attached TFS inside its housing |
| 4 | yes | yes/yes | "—" |
| 5 | yes | no/yes | PPt at TFS position inside TFS housing |

**Table 1**

List of experiments (Exp.) conducted with the PPt at various positions inside the UCASS and TFS housing during wind tunnel flow calibrations.

| fit | a | b |
|-----|---|---|
| (0°,0°) HRC | 1.040 | 0.216 |
| (10°,10°) | 0.995 | 0.119 |
| (10°, -10°) | 1.024 | 0.075 |
| (15°,20°) | 0.987 | 0.120 |
| (20°,5°) | 0.972 | 0.017 |
| (0°, 30°) | 0.910 | 0.073 |
| (0°, -30°) | 0.890 | 0.142 |

**Table 2**

Parameters of the linear fit function ($v_{PPt} = a \cdot v_{TA} + b$) to selected UCASS-internal flow velocities measured under variable angle of attack (AOA) in reference to the main wind tunnel flow speed.





| angle $(\varphi, \vartheta)$ | deviation $\Delta v_{\mathrm{rel}}^{\max}$ in % | deviation $\Delta \bar{v}_{\mathrm{rel}}$ in % |
|---|---|---|
| (0°,0°) | 1.1 | 0.6 |
| (0°,5°) | 1.7 | 1.0 |
| (0°,30°) | 3.5 | 1.8 |
| (0°, -30°) | 2.6 | 1.5 |
| (5°, -5°) | 1.5 | 1.0 |
| (10°,10°) | 2.8 | 0.9 |
| (10°, -10°) | 1.6 | 0.7 |
| (15°,20°) | 2.1 | 0.7 |
| (20°,5°) | 3.3 | 1.5 |
| (20°, -40°) | 8.6 | 4.9 |
| (30°,0°) | 7.3 | 4.1 |
| (35°, -20°) | 9.0 | 4.6 |
| (35°,40°) | 13.4 | 6.1 |
| (45°,15°) | 9.5 | 5.5 |
| (50°, -5°) | 10.6 | 6.7 |
| (0°,0°) | 1.4 | 1.0 |

**Table 3**
Maximum ($\Delta v_{\mathrm{rel}}^{\max}$) and mean values ($\Delta \bar{v}_{\mathrm{rel}}$) of the percentual relative deviation of measured flow
velocities under variable angle of attack (AOA) for the calibration curve of TFS 8 (cf. Fig. 5).


| angle $(\varphi, \vartheta)$ | deviation $\Delta v_{\mathrm{rel}}^{\max}$ in % | deviation $\Delta \bar{v}_{\mathrm{rel}}$ in % |
|---|---|---|
| (0°, 0°) | 1.8 | 0.8 |
| (0°, 5°) | 4.6 | 3.3 |
| (0°, 30°) | 16.5 | 15.0 |
| (0°, -30°) | 16.1 | 15.3 |
| (5°, -5°) | 3.9 | 2.5 |
| (10°, 10°) | 7.3 | 6.2 |
| (10°, -10°) | 6.0 | 4.5 |
| (15°, 20°) | 8.4 | 7.0 |
| (20°, 5°) | 12.3 | 10.5 |
| (20°, -40°) | 33.4 | 31.1 |
| (30°, 0°) | 27.7 | 24.9 |
| (35°, -20°) | 38.1 | 35.1 |
| (35°, 40°) | 54.5 | 51.8 |
| (45°, 15°) | 47.2 | 44.1 |
| (50°, -5°) | 57.5 | 56.1 |
| (0°, 0°) | 3.8 | 2.1 |

**Table 4**
Maximum ($\Delta v_{\mathrm{rel}}^{\max}$) and mean relative deviation ($\Delta \bar{v}_{\mathrm{rel}}$) of in-UCASS measured flow velocity versus
the wind tunnel flow speed as a function of the UCASS's angle of attack (AOA) in reference to
velocity ratio in zero-position (in correspondence to measurements shown in Fig. 6).




| $T_{amb}$ and | | 275 ± 0.3 K | 264 ± 0.3 K | 254 ± 0.3 K |
|---|---|---|---|---|
| corresponding | $h$ (Fig. 7a) | ~3930-3990 m | ~5700-5790 m | ~7250-7310 m |
| heights | $h$ (Fig. 7c) | ~3690-3740 m | ~5490-5560 m | ~7250-7320 m |
| in soundings | $h$ (Fig. 7e) | ~3700-3770 m | ~5400-5490 m | ~7230-7300 m |
| RS41 SGP | $\Delta p$ | | ± 1.0 hPa | |
| | $\Delta T$ | — | ± 0.3 K | — |
| | $\Delta h$ | | ± 10.0 gpm | |
| TFS | $v_{TFS}^{corr}$ (Fig. 7a) | 6.1 – 6.3 m s$^{-1}$ | 6.7 – 6.9 m s$^{-1}$ | 5.9 – 6.0 m s$^{-1}$ |
| | $v_{TFS}^{corr}$ (Fig. 7c) | 6.3 – 6.4 m s$^{-1}$ | 6.1 – 6.3 m s$^{-1}$ | 5.8 m s$^{-1}$ |
| | $v_{TFS}^{corr}$ (Fig. 7e) | 5.0 – 5.5 m s$^{-1}$ | 6.3 – 6.4 m s$^{-1}$ | 5.4 – 5.7 m s$^{-1}$ |
| | $\Delta v_{TFS}$ | ± 0.4 % | ± 0.4 % | ± 0.4 % |
| | $\Delta v_{TFS}^{corr}$ | ± 7.9 % | ± 8.8 % | ± 6.9 % |
| GPS | $v_{GPS}$ (Fig. 7a) | 6.7 m s$^{-1}$ | 6.7 m s$^{-1}$ | 7.6 m s$^{-1}$ |
| | $\Delta v_{GPS}$ | -8.8 – -6.0 % | \| 0.0 – 3.5 % \| | -29.3 – -27.7 % |
| | $v_{GPS}$ (Fig. 7c) | 6.6 m s$^{-1}$ | 6.4 m s$^{-1}$ | 6.5 m s$^{-1}$ |
| | $\Delta v_{GPS}$ | \| -4.4 – -2.9 % \| | \| -4.5 – -1.4 % \| | -14.2 – -12.5 % |
| | $v_{GPS}$ (Fig. 7e) | 5.7 m s$^{-1}$ | 7.1 m s$^{-1}$ | 7.0 m s$^{-1}$ |
| | $\Delta v_{GPS}$ | -14.1 – -4.2 % | -12.8 – -10.8 % | -30.8 – -23.2 % |

**Table 5**
Summarised uncertainties 1) $\Delta T$, $\Delta p$, and $\Delta h$ (GPS-altitude) from RS41 SGP radiosondes, 2) the
overall uncertainty $\Delta v_{TFS}^{corr}$ that result from Gaussian error propagation (Eq. 1 and Eq. 2) and that
combine uncertainties from the RS41 SGP data and conditions during TFS calibrations
(considering only $\Delta T$, $\Delta p$, $\Delta T_0$, and $\Delta p_0$) yielding $\Delta v_{TFS}$ and b) the uncertainty ($\sigma$) in $v_{TFS}$ obtained
from cold chamber experiments (electronic supplement, Sect. S 1). 3) For three selected ambient
temperature conditions ($T_{amb}$ of 275 K, 264 K, and 254 K, from cold chamber experiments) the
resulting deviations ($\Delta v_{GPS}$) of absolute $v_{GPS}$ in reference to absolute $v_{TFS}^{corr}$ from three vertical
soundings (see Fig. 7 and adjacent text) are provided for comparison.

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
