# Peer review of "Improving the accuracy in particle concentration measurements of a"

_EGUsphere, 2025_

## Author Response (AR1)

**Authors' replies to the reviews of the ACP manuscript**

On behalf of all authors, I would like to express my gratitude and appreciation to the two reviewers for their valuable suggestions and constructive critiques, which contributed decisively to optimising and completing the presented study. We hope to have adequately addressed all comments and objections and hereby submit a revised version of the article for re-evaluation and thank the reviewers in advance for their renewed efforts.

[RC1] Please rephrase the following lines for clarity and readability:

- o Lines **41-42**
- o Lines **473-475**
- o Lines **476-477**
- o Lines **502-509**

[AC] We have followed the reviewer's suggestions and revised the passages indicated.

[RC1]: Please correct the following typos:

- o Lines **281-282**: Change to **"Figure 4"**

**[AC]: corrected**

- o Line **394**: Change to **"percentile"**

**[AC]: corrected into „percentage"**

- o Line **474**: Change to **"height"**

**[AC]: corrected as suggested**

[RC1]: In **Section S 4**, please rephrase the last sentence starting with **"Despite..."** for better clarity.

[AC] We have followed the reviewer's suggestions and rephrased this sentence.

[RC2]: The heat flow sensors (TFS) are of crucial importance for this study. Nevertheless, technical details about the applicability for measurements in the UTLS were not mentioned. Please add a description of the ability of these sensors to operate at temperatures lower than -20°C (= lowest temperature of the cold chamber test).

> [AC]: We understand the reviewer's request for performance information of the TFS to altitudes beyond the focussed 7.5 km. However, the scope of this article is limited to the troposphere up to specified altitude, where also the -20°C level is reached (corresponding to the lowest temperature of the cold chamber test and the temperature range specified by the TFS manufacturer). To a certain point, the inclusion of requested information would hardly fit to the content of the current article. Moreover, the majority of UCASS particle detections happened at heights below 7.5 km, hence, the altitude range chosen still covers the most relevant part of the flight in terms of particle observations. Investigations concerning the capability of the thermal

flow sensors to perform also at temperatures below -20°C may be part of future work. However, for clarification, in the text throughout the article the vertical limit set and thus the main scope of this article is better emphasised.

[RC2]: It is not clear to me why 19 different (?) sensors were used for this study as Table S1 suggests. Are there any (technical) differences between the sensors? Furthermore, the nomenclature of the TFS is not consistent throughout the manuscript (TFS A/B in Line 464; TFS 7/8 in Lines 407-408; TFS°1/2 in Lines 451-452). Again, what are the differences here? Please add more information and change to a consistent nomenclature which also makes it easier for the reader.

[AC]: Thank you for drawing our attention to the ambiguities in the naming of the TFS. We have substantially shortened Table S1 to focus only on the calibration results shown in the paper. The naming of the TFS in Section 5 for the balloon soundings has been consistently renamed to A and B, which has also been adopted for the revised Figures 7 and S10. With added cross-references to Tables S1 and S3 and explanations in the text, we hope to meet the reviewer's request.

[RC2]: In the abstract it says "In-flight comparisons revealed that UCASS sample flows rarely match the balloon's ascent rate, instead, equality (vGPS = vTFS) is achieved only at AOA ≠ 0°, potentially affecting the UCASS-internal flow pattern and particle transmission efficiency." which I would consider as a major finding of this study. If I am not mistaken, this aspect is extensively discussed in Section 5.1, but not explicitly measured or investigated (AOA was not explicitly mentioned in Section 5). Furthermore, Figure 7 shows the deviations in velocities at different altitudes, but neither does it show the angle of attack. Can more information about the wind conditions during the ambient measurements be provided here? For example, can additional information on wind velocity be included in Figure 7?

[AC]: The quantitative impact of the AOA on the flow velocities ratio (inside/around UCASS) is derived from the laboratory experiments only. During the atmospheric soundings, the AOA information is usually not available. Along with the revision of the article, we have clarified this in the abstract. The velocity ratios measured strongly indicate an AOA ≠ 0. From one individual flight and based on the inclination of the total setup (balloon - payload ensemble) we found that deviations of the UCASS from an isoaxial sampling are likely to occur in general. The exact conditions and perturbances potentially influencing the flow conditions through the UCASS are still not completely understood, which, however would be essential to know for correcting GPS-based flow rates. Future work on this issue may provide further insight concerning the impact of pendulum and rotation moments or a superposition of both on the UCASS measurements. In any case the direct measurement of the sample flow through UCASS at any time during flight circumvents the consideration of all these effects. Correspondingly to the reviewer's request, we added observational data of windspeed and winddirection to Fig. 7 and provided additional information to the text.

[RC2]:Line 332 + Line 335: The terms HRC and TPC were introduced as if they were common knowledge, which they are not for me (and probably for many other readers). What are these types of calibrations and why were they used for this specific issue? Why does the HRC only offer a "relatively" high-resolution (as written in Line 332)? Why is the TPC "robust" as stated in the conclusion? Please add the relevant information.

[AC]: We have followed the reviewer's suggestions and added relevant information and explained the abbreviations HRC and TPC. The word "relatively" referred to the comparison with the TPC and has now been removed from the text.

[RC2]: General: Please follow the AMT Submission-Guideline "For items other than units of time or measure, use words for cardinal numbers less than 10; use numerals for 10 and above." throughout the manuscript (e.g. Line 232: fifteen velocity measurements).

[AC]: We have followed the reviewer's suggestions and made the corresponding changes to the text.

[RC2]: Line 235: I don't see any horizontal bars in Figure 3.

[AC]: We have changed Figure 3 and added further information to the text and the caption of the figure.

[RC2]: Line 281: Figure 4

[AC]: corrected as suggested

[RC2]: Code availability: This will be the software package used for the fits, but does not describe whether and how the code or script used for this study is/will be made available.

[AC]: Both data and the program code are available in a repository via "https://doi.org/10.5281/zenodo.15519552". This information has also been included in the main article.

We have made some additional, minor changes to the text, which can be seen in the change track.

---

## Author Response (AR2)

**Authors' replies to the reviews of the ACP manuscript**

On behalf of all authors, I would like to express my gratitude and appreciation to the reviewers and editor for their valuable suggestions and constructive critiques. We hope to have adequately addressed the final comments and hereby submit a revised version of the article and thank the reviewers and editor in advance for their renewed efforts.

[RC] Why was the naming of the TFS in section 5 changed from numbers to letters? I would have stayed with numbers for a consistent nomenclature, especially to avoid possible confusion with the calibration coefficients A and B (for example in Table S3).

> [AC] We have followed the reviewer's suggestion and changed the naming of the TFS to numbers to avoid confusion.

We have also checked the formatting of the references so that it complies with the Copernicus guidelines. We have taken into account that shading in tables (Table 5) cannot be used in the final accepted manuscript and have changed this.